# An Enhanced File Transfer Mechanism Using an Additional Blocking Communication Channel and Thread for IoT Environments

**DOI:** 10.3390/s19061271

**Published:** 2019-03-13

**Authors:** Yangchan Moon, Mingyu Lim

**Affiliations:** 1Department of Internet & Multimedia Engineering, Konkuk University, 120 Neungdong-ro, Gwangjin-gu, Seoul 05029, Korea; myc1051@konkuk.ac.kr; 2Department of Smart ICT Convergence, Konkuk University, 120 Neungdong-ro, Gwangjin-gu, Seoul 05029, Korea

**Keywords:** file transfer, non-blocking communication channel, blocking communication channel, communication framework

## Abstract

In this paper, we propose an enhanced file transfer mechanism for a communication framework (CM) for Internet of Things (IoT) applications. Our previous file transfer method uses a basic non-blocking communication channel and thread for the CM (non-blocking method), but this method has a cost of adding additional bytes to each original file block. Therefore, it is not suitable for the transfer of large-sized files. Other existing file transfer methods use a separate channel to transfer large-sized files. However, the creation of a separate channel increases the total transmission delay as the transfer frequency increases. The proposed method uses a dedicated blocking communication channel in a separate thread (blocking method). The blocking method uses a separate channel and thread which are dedicated to transferring file blocks. As it creates the separate channel in advance before the file transfer task, the proposed method does not have an additional channel creation cost at the moment of the file transfer. Through file transfer experiments, the blocking method showed a shorter file transfer time than the non-blocking method, and the transmission delay was increased as the file size grew. By supporting both non-blocking and blocking methods, an application can flexibly select the desirable method according to its requirement. If the application requires the transfer of small-sized files infrequently, it can use the non-blocking method. If the application needs to transfer small-sized or large-sized files frequently, a good alternative is to use the blocking method.

## 1. Introduction

Many of today’s multiuser-based distributed applications such as online games, social networking systems (SNS), and file sharing systems provide various communication functions for efficient interaction among users. We developed a communication framework called CMSNS [1] especially designed for SNS environments. Using CMSNS, a developer can easily build an SNS because CMSNS provides high-level communication functionalities such as user management and content management. Due to the high-level communication services, the developer does not need to implement such functionalities from scratch. Among various communication functionalities, file transfer is an important and frequently used function.

We also expect that Internet of Things (IoT) networks will become similar to SNS networks, because frequent and random transfer of small-sized and large-sized files may be required in a future IoT environment where a variety of devices capture and transfer not only small sensor data files but also large data files like sensed images [2]. The emergence of social and opportunistic IoT concepts [3] reflects such a trend, too.

Traditional standard file transfer protocol (FTP) [4,5], other existing file transfer protocols [6,7,8,9], and systems that have developed file transfer functions [10,11] have been focusing on a method for rapidly transferring large-scale files. In addition, applications that share content among users such as SNS require a file transfer function, and a communication framework or communication middleware that provides various communication functions including the file transfer service have also been studied [12,13,14,15,16].

However, existing file transfer methods specialized for large-scale file transfer are not suitable for transferring relatively small-sized files at a high frequency because each file transfer requires the creation of additional communication channels which are separate from the control channel. In order to create a socket for Transmission Control Protocol (TCP), which is a transport layer protocol used for file transmission, a delay of several hundred milliseconds to several seconds may occur depending on the network conditions. Therefore, the communication channel connection setup time may affect the total delay of frequent file transfer.

Our previous file transfer mechanism of CMSNS, which is called the non-blocking method, uses a control channel (a basic non-blocking communication channel) to transfer a file, and it does not need the cost of additional channel management. However, it has a burden of piggybacking each file block on a predefined CM event, which makes the size of transferred bytes larger than that of the original file block. Therefore, the non-blocking method is suitable only for transmitting small-sized files with a low frequency.

An alternative to using TCP for file transfer is to use the User Datagram Protocol (UDP) [17]. Although UDP is more lightweight and faster than TCP, a crucial drawback is that it is unreliable. As TCP and UDP have their own advantages and disadvantages, we chose to use TCP as the transport layer protocol because it is more common in existing methods. In addition, recent IoT trends and industry needs favor TCP. [18]

In order to solve the problem of the existing file transfer methods described above, this paper proposes an enhanced file transfer mechanism for our communication framework (CM). The enhanced file transfer method, which is called the blocking method, uses a dedicated blocking communication channel and a thread. In the blocking method, a dedicated communication channel and a thread are used for file transfer similar to the conventional FTP style. The transmission of a CM event required for the preprocessing and postprocessing of the file transfer between the transmitting node and the receiving node uses the basic non-blocking channel, and the transmission of the file block generates and uses the dedicated blocking channel and the thread separately. This method is suitable for frequently transferring small-sized or large-sized files because the file block is directly transferred without being converted into a CM event. To overcome the problem of separate channel creation cost, in the blocking method, when a client logs in to a server, a dedicated channel for file transmission is created in advance to reduce the channel generation delay time during file transmission.

Experimental results show that the blocking method has a shorter file transfer time than the non-blocking method. The non-blocking method has a longer transmission time because of the cost of converting a file block into a CM event. The transmission time difference was not significant in the transfer of small-sized files, but the transmission delay increased as the file size grew. The increase of the total transmission delay becomes worse if file transfer at a high frequency is needed. As a result, CM supports both the non-blocking and blocking methods so that an application can flexibly choose a desirable file transfer service according to its own requirements. If an application requires the transfer of small-sized files infrequently, it can use the non-blocking method. If the application needs to transfer small-sized or large-sized files frequently, a good alternative is to use the blocking method.

The remainder of this paper is organized as follows. In Section 2, we introduce our communication framework and previous file transfer method, which is a basis of the proposed file transfer mechanism. In Section 3, we describe in detail the procedure of the proposed file transfer method. In Section 4, we compare the transmission times of the two file transfer methods according to different file sizes and geographical locations of the sending and receiving nodes. After an analysis of existing file transfer approaches in Section 5, the paper is concluded in Section 6.

## 2. Communication Framework (CM)

CM [1] is a communication framework for developing distributed applications. Using the API and configuration files provided by CM, an application developer can easily implement various communication functions that enable the application to interact with other CM nodes (applications that use CM communication services). Using CM communication services, a CM node can distinguish roles from client to server. The CM client node can easily interact with the CM server node with basic communication functions such as registration, connection, login, CM event creation and transmission, CM event reception and processing, etc. In addition, the CM node can use services such as file transfer, SNS content management, diagnosis of the current network status, and communication channel management. All CM services used by the client are configured in a request–response pattern. When a client requests a service, the client CM creates the request event and sends it to the server. When the server performs the requested service, the server CM creates a response event and sends it back to the client.

### 2.1. Event-Based Asynchronous Communication

Existing CM nodes communicate internally in an event-based asynchronous manner. The advantage of the asynchronous communication method compared to the synchronous method is that it does not have to wait for the response event to the request event. Accordingly, the CM application can perform other tasks after requesting the communication service of the CM without waiting for a response. Instead, the application registers a callback method (or function) dedicated to receiving an event in CM so that the application can receive a response to the previous request from CM at any time during the execution of another task. Because CM invokes the registered callback method each time it receives an event, the application can process the response event received at any instant after the CM service request is sent.

To support these event-based asynchronous communication schemes, CM runs with multithreads (main thread, processing thread, sending thread, receiving thread) as shown in Figure 1. First, when the main thread of the application runs CM, it starts a processing thread, a sending thread, and a receiving thread in CM. The main thread is responsible for processing local events from application users. The processing thread is responsible for receiving and processing CM events received from the receiving thread. The processing of a CM event is divided into two cases: the event processing in the underlying CM layer and that in the application layer. In order for an application to process a CM event, it registers an event handler object including an event processing method in CM. The processing thread may process the received CM event internally first, if necessary, and then call the event processing method of the application event handler if the application needs to process the event, too.

The sending thread is responsible for converting a CM event created by the main thread or the processing thread into a low-level byte message and transmitting it to the network. When the main thread or the processing thread needs to send a message to a remote node, it creates a necessary CM event, puts it in the sending queue, and forwards it to the sending thread. The receiving thread is responsible for receiving a byte message from the network, converting it into a CM event, and passing the event to the processing thread. The receiving thread puts the received CM event into the receiving queue, and the processing thread receives and processes the received CM event.

### 2.2. File Transfer Mechanism (Non-Blocking Method)

Figure 2 shows the concept of the non-blocking method. The non-blocking socket channel, which is the default communication channel of CM, is the channel that CM clients use by default when connecting to the CM server. The non-blocking channel is registered in the selector object of CM, and the CM receiving thread detects and receives a CM event delivered asynchronously by monitoring the selector object. Since all messages sent and received through the selector follow the CM event format, when CM receives a byte message over the channel, it converts the message into a CM event and processes it. When CM sends a message, it goes through the opposite process. Therefore, in order to transmit a file using the non-blocking channel, a file block cannot be transmitted as it is and must be converted to the CM event format. A CM event consists of two parts: an event header common to all events and an event body that has different payload according to an event type and identifier. As CM defines an event for the file block transfer, the file block can be included in the body part of this event and can be transmitted.

Table 1 summarizes the protocol of the non-blocking method when a receiver node requests a file from a sender node. In the protocol, all the preprocessing steps (1–4), the file transfer step (5), and the postprocessing steps (6,7) use only the default non-blocking channel. Since every event contains information about the currently transferred file as a separate field value, CM can process it asynchronously (independently) even if other events are interleaved during the file transfer process. For example, even during the file transfer process, a client, which is a receiving node, can request and receive another service from a server that is a transmitting node.

The non-blocking method does not need the management of a separate data channel for the file transfer. Instead, whenever a file block is transmitted, it must be configured as a CM event each time, which can result in a relatively high transmission cost. Therefore, this method can be considered a suitable method for transferring a small-sized file at a low frequency. Through a performance evaluation, we analyze the advantages and disadvantages quantitatively herein by comparing the non-blocking method and the proposed blocking method.

## 3. Enhanced File Transfer Mechanism (Blocking Method)

In the previous non-blocking method, CM can transfer a file using the existing channel without generating a separate channel. However, this has the cost of converting all file blocks into CM events. In this section, we describe the process of performing a file transfer on a separate thread using a dedicated blocking channel. As shown in Figure 3, a separate blocking channel is a dedicated channel for a sending node to synchronously transmit file blocks to a receiving node, without an additional conversion process. A blocking socket channel can be added through a channel service of CM. The channel creation time can be considered in two ways. One way used by the existing FTP-style approaches is to add a blocking channel only when the file transfer is requested. In this method, there is no need to maintain additional channels until the file transfer service is used, but the transfer time may become longer because the channel must be added at the time of file transfer. To solve the problem of the additional cost of channel creation during the file transfer, another method is to create a channel in advance when the CM client logs in to the CM server. This method used by the blocking method should keep additional channels even if file transfer is not used, but it is possible to transfer a file faster because there is no need to create an additional channel when starting the file transfer task. For the synchronous transmission of file blocks, a separate thread handles the transmission task so that the sending node and the receiving node can also send and receive different events in parallel. This dedicated thread is created only when the file transfer task starts. To avoid the cost of thread creation, CM gets a new thread from a thread pool that creates and manages available threads in advance. While CM uses the default non-blocking channel to exchange control events required for the preparation preprocessing and the postprocessing tasks before/after the file transfer, it uses the separate blocking channel and thread to transfer file blocks.

We describe a detailed file transfer process of the blocking method as an example of a file reception service (*requestFile()*). In this example, it is also assumed that the client is a receiving node that requests a file and that the server is a sending node that transfers the file. The file transfer manager handles the request and the process of a file transfer. Table 2 shows the protocol of the blocking method between the sending and the receiving nodes.

In Steps (1)–(4), the sending and the receiving nodes perform a necessary synchronization preprocessing through the exchange of control events before starting the file transmission. In Step (5), the sending node transmits file blocks to the receiving node. For this step, both the sending node and the receiving node use a separate thread and a blocking channel dedicated to the file transfer task. Steps (6) and (7) are the postprocessing in which the sending and the receiving nodes perform a finishing task after the file transmission is completed.
When the client application calls the *requestFile()* method of CM, the file transfer manager of the client CM sends a corresponding request event to the server via the non-blocking channel.The server CM receives the request event asynchronously, and the file transfer manager then sends a response event that includes a field value indicating whether file transfer is possible. The client CM receives the response event as a result of the file transfer request. If the request is rejected, the client’s file transfer request is terminated upon receipt of this response event.If the requested file transfer service is available, the server CM sends a control event to the client to notify it of beginning the file transfer and stores the file information in the sending file list. When the client CM receives this event, it searches for the blocking channel with the server to check the connection status. The client CM adds the file information to be received to the receiving file list. Then, the client CM requests through the thread pool to perform the file receiving operation in a separate thread. After the file-receiving thread opens the file, it waits for receiving file blocks through the blocking channel.The client CM sends a response event to the server to indicate that it is ready to receive the file.When the server CM receives the response event from the client, it requests the file transmission task through the thread pool to proceed via the blocking channel in a separate thread. The file-sending thread of the server CM opens the file, reads the file data in block units, and transmits the file data to the client through the blocking channel.When the file-sending thread of the server CM sends all file blocks, it sends a transfer-complete event to the client via the default non-blocking channel. Lastly, the file-sending thread closes the file that was open and exits.When the client CM receives the transfer-complete event, it waits until the file-receiving thread receives all the file blocks and terminates, and it deletes the received file information from the receiving file list. The client CM then sends a response event to the server and completes the file reception operation. When the server CM receives the response event from the client, it deletes the sent file information from the sending file list and completes the file transmission operation.

Comparing file transfer with that using the non-blocking method, the blocking method does not need to convert it into a CM event when a file block is transmitted. However, a separate channel must be added to the default channel for file block transfer. Therefore, this method is suitable for transmitting small files at a high frequency or for transmitting large-sized files even if the transmission frequency is low. More detailed features of the blocking method were analyzed by a separate quantitative performance evaluation in comparison with the non-blocking method.

The file transfer service in CM supports the non-blocking and the blocking methods. The two file transfer methods can be selected through the *FILE_TRANSFER_SCHEME* field value in the CM server’s configuration file (cm-server.conf). If the field value is 0, the non-blocking method is used. If the field value is 1, the new blocking method is used. The CM client receives the information on the current file transfer method from the server when logging in to the CM server and uses the same file transfer method as the server. A further option in the server configuration file for the CM’s file transfer service is the append/overwrite mode. The CM server can select the appropriate option in the *FILE_APPEND_SCHEME* field. If the value of this field is 0, the overwrite mode is set. If the value is 1, the append mode is set. In the overwrite mode, if the same file as the file name to be transferred exists in the file receiving node, the existing file is deleted and all the entire file blocks are transmitted again. In the append mode, if a file with the same file name exists in the receiving node, the transmitting node transmits only the remaining file blocks excluding the existing file blocks. The file receiving node may dynamically select these two modes when requesting a file. The CM’s file transfer service can be used by applications calling the *pushFile()* or *requestFile()* method. The *pushFile()* method is a service that sends a file to a target node, and the *requestFile()* method is a service that requests a file to be received from a source node.

## 4. Experimental Results

In this section, we compare and analyze the performance of the non-blocking method and the blocking method provided by CM through the measurement of transmission time. The experimental environment is as follows.

For the experiment, we developed a sample client and server applications using CM with Eclipse IDE and Java. The computer that ran the server (Windows 10, i3 3.5 GHz, 8 GB memory) was connected to 1 Gbps wired LAN and the client computer (Mac OS, i5 2.9 GHz, 8 GB memory) located in different places was connected to the server via wireless LAN (WiFi). The location of the client was configured in the same laboratory, the same campus as that of the server, and in a different city near to that of the server. The client measured the file transmission delay while receiving files of various sizes from the server through the CM’s file transfer service (*requestFile()*). The file transfer time was measured as the total elapsed time from when the client requested the file transfer to when the file transfer was completed. The size of the files used for the transfer varied from 1 kilobyte (KB) to 100 megabytes (MB), and they were all image files. To measure the transmission time of the entire file, the value of the *FILE_APPEND_SCHEME* field of the configuration file in the server CM was set to 0, that is, the overwrite mode. The transmission time of each file represents the average of 10 transmissions. Figure 4, Figure 5 and Figure 6 show the measured file transfer time according to the physical distance between the client and the server.

The analysis results of the experiments of the file transfer time measurement are as follows. In general, regardless of the physical distance between the server and the client, the non-blocking method has a longer transmission time than the blocking method. This is because the non-blocking method has an additional cost when converting the file format to the CM event format. The transmission time difference of the two methods varies from several milliseconds to several hundreds of milliseconds depending on the size of the file used in the experiment. When transferring small files as small as 10 KB, the difference between the two methods is only a few milliseconds. When transferring large files of 100 MB or more, there are differences of hundreds of milliseconds. That is, as the file size increases, the file transfer time does, too.

The blocking method shows shorter transfer times than the non-blocking method. However, this method requires the use of a separate communication channel rather than the CM’s basic channel for the file transfer. In this experiment, since the blocking channel was previously created and maintained when the client logged in to the server, the channel addition cost was not included in the file transfer time. If all clients create additional channels in advance for file transfer, the CM has a burden of managing twice as many channels as usual, which may be a problem for scalability of the system. If a blocking channel is added only when transmitting a file, like in most existing file transfer methods [4,5,6,7,8,9,10,11,12,13,14,15,16], including the well-known FTP, the channel management cost can be reduced, but the file transmission time is increased by the channel addition time. Figure 7 shows the result of measuring the time taken by the client to add socket channels and complete synchronization with the server. CM’s channel addition service can be used synchronously or asynchronously, and we measured the server response time for both cases. As shown in the figure, the channel addition is a high-cost operation that can take up to several seconds. In fact, the channel add-on time varies considerably from a few hundred milliseconds to a few seconds depending on the situation. As a result, if a channel is added when there is a file transfer request, the time required for the entire file transfer may be greatly increased, resulting in a slower file transfer performance than the non-blocking method.

## 5. Related Work

The File Transfer Protocol (FTP) is a representative standard protocol [4,5]. Most commercial file transfer clients and servers are implemented according to this standard specification for compatibility. In the file transfer procedure, the blocking method is similar to the FTP method in that the control message channel and the file data channel are separately managed. However, FTP does not describe the consideration of the channel type or the thread usage, which should be considered when implementing a file transfer process as a service of a communication framework rather than as an independent file transfer program.

GridFTP [6,7], an extension of existing FTP, was developed to control TCP buffer size and use multiple channels for secure, reliable, and high-performance data transmission. In other words, GridFTP was designed as a useful protocol for the transfer of large-sized files. However, in order to transmit a small number of files having a small size, it still requires separate channel management and does not consider the relevant cost.

Bittorrent [8] is a file transfer method that significantly reduces the transmission time by receiving file blocks from multiple source nodes on a peer-to-peer basis in order to efficiently transfer large-scale files. This method is mainly used for large file transfers such as of video files. On the other hand, when a small-sized file is transmitted, the receiving node still receives the source node information through the tracker node, obtains the torrent file, and receives the file blocks from the source, which requires longer transmission time than file transfer from a known, single source node directly.

Dual-Direction FTP (DDFTP) technology [9] has been proposed as a way to transfer large files quickly by using multiple replicated FTP servers. DDFTP uses a method for specifying file blocks to be transferred among servers in order to transfer file blocks in parallel from multiple servers. This technique also focuses on how to transfer large files quickly, but for simple file transfer, it still requires synchronization between client and server and separate channel and thread management.

A recent file transfer study [10] proposed a file transfer method that can synchronize multiple files among various heterogeneous devices like N-Screen in a short time. The existing FTP needs to connect the data channel repeatedly every time a file is transferred. To solve this problem, this research used a method to keep the data channel connected. This method is similar to the blocking method proposed in this paper. However, even if small-sized, low-frequency file transfer is required, this research still maintains unnecessary channels, which may cause resource waste.

Fan’s research [11] proposed the FTP-NDN technology, which is a file transfer method applying a peer–peer-based encryption scheme in a Named Data Network (NDN) environment. The main purpose of this technique is not to receive a file only from a fixed source node, but to find a close node with the same file and to transmit the file in an encrypted form. As this research focuses on encryption, it does not describe the details of communication channel management.

Communication frameworks or middleware for online social networks [12,13,14,15,16] provide a variety of communication services for applications, among which image file transfer is a basic function. However, since these studies focus on efficient interaction services among users, there is no specific description of file transfer procedure.

In summary, existing research on file transfer mainly focuses on methods for the fast transfer of large-scale files. However, they did not consider the cost of managing the separate data channels necessary for the file transfer or simplification of the file transfer process for small-sized files.

Bormann et al. [17] devised block-wise transfers in the Constrained Application Protocol (CoAP). While the above other existing methods use TCP as their transport layer protocol, CoAP uses UDP because of its simplicity. However, a sender CoAP node must wait for an acknowledgement (ACK) message for each transmitted block to check the delivery was successful, and it should retransmit the block if the previous transmission fails.

## 6. Conclusions

In this paper, we proposed a communication framework (CM) that selectively provides a file transfer service according to the requirements of IoT applications. The CM file transfer service consists of two methods: non-blocking and blocking methods. The application can flexibly choose between the two methods to meet its requirements. The non-blocking method is suitable for an application that transmits small-sized files with a low frequency, since it does not have a separate channel management cost but the method does increase the cost of file block transmission. The blocking method is suitable for applications that transmit large-sized files or small-sized files at a high frequency because the file blocks can be transmitted quickly, although this method has increased additional channel and thread management costs.

Currently, two file transfer methods supported by CM can be specified through the configuration file of the server CM. That is, when the server selects the file transfer method, the clients are also bound to use the same file transfer service. In future work, we plan to complement the two sending and receiving nodes to dynamically select two file transfer services. In addition, in the file transfer experiment of this paper, we assumed that the sending server does not interact with other clients except the receiving node. In the next experiment, we will conduct file transfer experiments in a more complicated situation that emulates real IoT environments. As the number and the load of the sending and receiving nodes are increased by adding CM servers and clients to randomly send and receive files of different sizes and frequencies, our mission is to find an optimal file transfer method to provide a small transmission time while minimizing the channel management cost for various IoT environments.

## Figures and Tables

**Figure 1 sensors-19-01271-f001:**
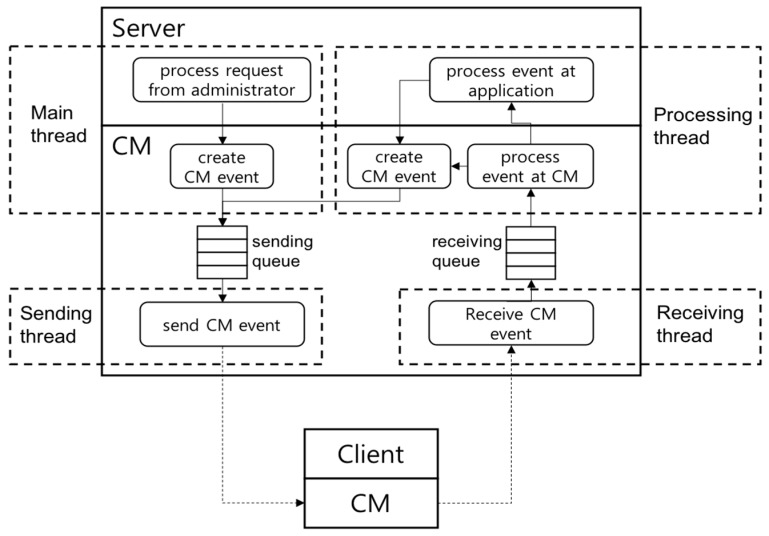
Communication framework (CM) multithreaded architecture.

**Figure 2 sensors-19-01271-f002:**
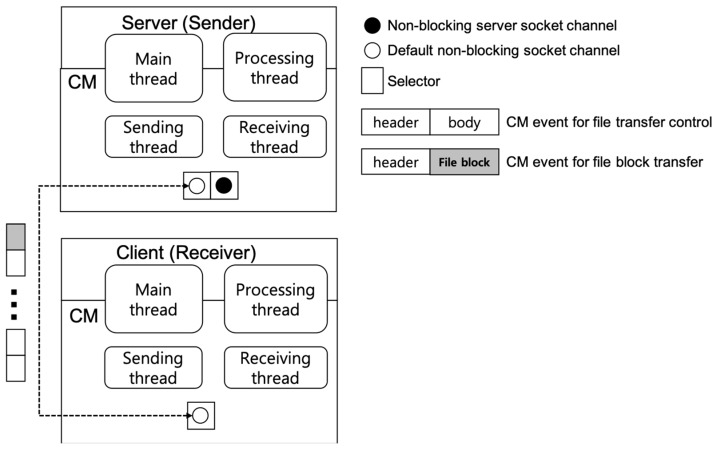
Concept of the non-blocking method.

**Figure 3 sensors-19-01271-f003:**
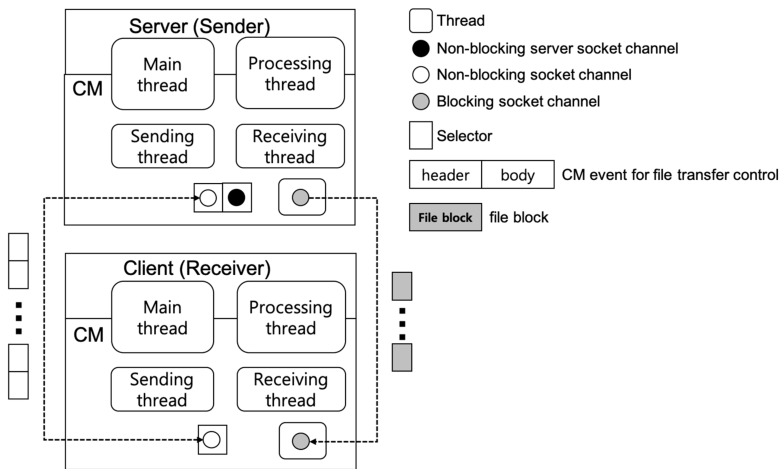
Concept of the blocking method.

**Figure 4 sensors-19-01271-f004:**
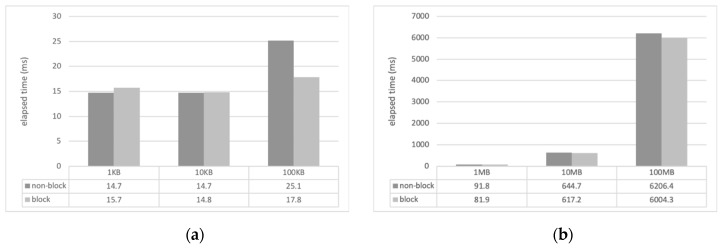
Comparison of file transfer time (server and client in the same laboratory): (**a**) small files in kilobytes; (**b**) large files in megabytes.

**Figure 5 sensors-19-01271-f005:**
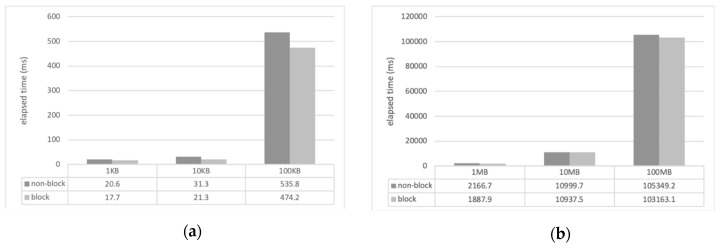
Comparison of file transfer time (server and client in different laboratories and on the same campus): (**a**) small files in kilobytes; (**b**) large files in megabytes.

**Figure 6 sensors-19-01271-f006:**
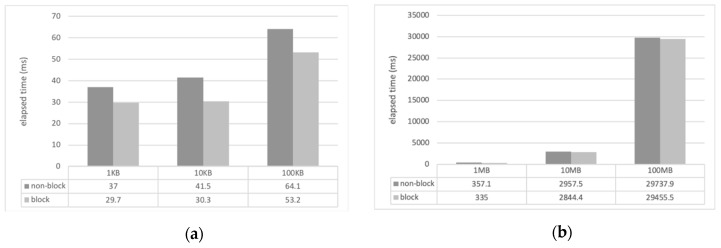
Comparison of file transfer time (server and client in different cities): (**a**) small files in kilobytes; (**b**) large files in megabytes.

**Figure 7 sensors-19-01271-f007:**
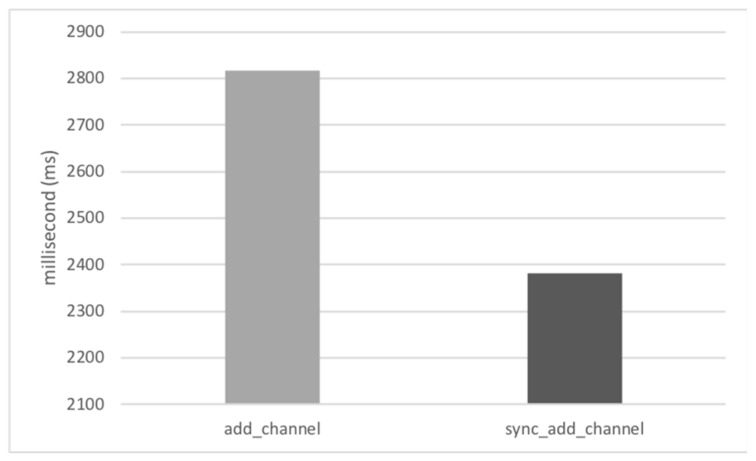
Elapsed time for adding a socket channel and synchronizing between communicating nodes.

**Table 1 sensors-19-01271-t001:** Non-blocking method protocol.

Step	Event Direction	Communication Channel	Description
(1)	Receiver→Sender	Non-blocking channel	Request file transfer
(2)	Sender→Receiver	Non-blocking channel	Reply file availability
(3)	Sender→Receiver	Non-blocking channel	Notify to begin file transfer
(4)	Receiver→Sender	Non-blocking channel	Complete to prepare file reception
(5)	Sender→Receiver	Non-blocking channel	Send file blocks
(6)	Sender→Receiver	Non-blocking channel	Notify to complete file transfer
(7)	Receiver→Sender	Non-blocking channel	Reply to complete file reception

**Table 2 sensors-19-01271-t002:** Blocking method protocol.

Step	Event Direction	Communication Channel	Description
(1)	Receiver→Sender	Non-blocking channel	Request file transfer
(2)	Sender→Receiver	Non-blocking channel	Reply file availability
(3)	Sender→Receiver	Non-blocking channel	Notify to begin file transfer
(4)	Receiver→Sender	Non-blocking channel	Complete to prepare file reception
(5)	Sender→Receiver	Blocking channel	Send file blocks
(6)	Sender→Receiver	Non-blocking channel	Notify to complete file transfer
(7)	Receiver→Sender	Non-blocking channel	Reply to complete file reception

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
