# Peer review of "An Enhanced File Transfer Mechanism Using an Additional Blocking Communication Channel and Thread for IoT Environments"

_sensors, 2019, doi:10.3390/s19061271_

Reviewer 1 Report

Overall comments

This paper proposes two new file transfer methods suitable for IoT environment, where small-sized files frequently made and shared. It look to be reasonably assumed that the preparation time of a file transfer will be longer than the time of the actual file transfer. (The reviewer thinks this is a kind of assumption because the argument does not have any reference.) Thus, the authors insist a new file transfer method should be designed since the traditional file transfer method such as FTP was designed to usually deal with a big-sized file. This is a right claim of problems. However, this paper is not good to be published in this journal due to a few of following reasons

1. The writing is too poor for readers to understand the paper.

Example 1:
The paper is wrongly titled with “File Transfer Mechanisms Using Blocking and Non-blocking Communication Channels for IoT Environments”

i)  When the reviewer read the title at first, the reviewer thought this paper was proposing two general file transfer methods applicable for any IoT environment. However, the actual contribution was the specific extended implementation of file transfer in the reference [14]. Then, to avoid any confusion, the paper title must be rephrased and limited with reflection of the actual points.

Example 2:
The misunderstanding becomes severe with the Introduction due to the following points.

i)  The paragraph organization is not proper. The first paragraph should be separated at the sentence “A file transfer method” on the line 39. The reviewer can observe the similar paragraph organization at the line 56. Since a single paragraph contains a few of key concepts together, the chained understanding is so difficult

ii)  The second paragraph beginning at the line 49 does not have any mention of the reference of [14]. Furthermore, the name of the specific system in [14] is named with a general noun phrase. At the line 61, “the specific CM event format” is suddenly appeared. All of these lead the reader to misunderstand.

iii)  The expression is not clear. For example, at the line 43, “small-sized files at a low frequency” is expected to be “small-sized files at a high frequency”. If it is true, it means small sized files are rarely transferred. Even though the cost of file transfer is high, the final disadvantage will be marginal.

Example 3:
The introduction does not have any summary of the contributions and papers organization. Suddenly, the chapter 2 starts with the explanation of the CM in [14]. After reading the chapter 2, the reviewer realized the meaning of the CM mentioned in the Introduction.

 The reviewer strongly suggests to reorganize and rewrite the paper from the begining in order to clearly deliver the contributions of the paper. Due to the increased interests of edge computing, small-sized file sharing system of IoT becomes more important issue. The reviewer think this paper can tell one of the solution of the issue.

Author Response

Attached please find the response file.

Reviewer 2 Report

Comparaison to the well known protocol such as FTP should be appreciated. That helps to evaluate the performance comparing to existing protocol.

Author Response

Attached please find the response file.

Reviewer 3 Report

The paper proposes a file transfer mechanism that can be exploited to transfer files between servers and clients. Specifically, two different mechanisms are presented: a blocking method that exploits two separate channels (similarly to FTP) and a non-blocking method that exploits a single communication channel. 

Even if interesting the authors presented the idea under the IoT umbrella, however, I cannot find almost anything which can be considered IoT-like. Specifically, IoT is, most of the time, characterized by devices with limited computational power that cannot handle TCP sockets, while the authors propose to exploit multiple sockets to handle data transmission. Furthermore, the authors also claim to exploit multiple threads both on clients and servers which is quite always unfeasible in the IoT domain. 

I cannot find a real novelty in the proposed work, a reader may argue that there are many different protocols that can be exploited to transfer files in the IoT domain. As an example, files can be transferred by exploiting CoAP along with the Blockwise option. In such way, there exists a single communication channel and the underlying layer will be UDP (more lightweight). 

Results are also not in-line with an IoT environment, in my opinion. Considering that, I think the paper is not ready to be published.

Minor comments:

- I think that the description of every single iteration following Figure 3, 4, 5 an 6 should be shortened. It is difficult to follow and it does not have a real contribution to the paper.

- Results measurements should be put into perspective. In IoT, even files of 1 MB are unusual.

Author Response

Attached please find the response file.

Round  2

Reviewer 1 Report

Please refer to the attached file

Author Response

Please refer to the attached response file.

Round  3

Reviewer 1 Report

Because the revised draft resolves all the questions of the reivewers, the reviewer thinks the draft deserves to publish.